# Systemic Anti-Inflammatory Effects of Intravenous Lidocaine in Surgical Patients: A Systematic Review and Meta-Analysis

**DOI:** 10.3390/jcm12113772

**Published:** 2023-05-31

**Authors:** Irene Castro, Pedro Carvalho, Nuno Vale, Teresa Monjardino, Joana Mourão

**Affiliations:** 1Department of Anesthesiology and Intensive Care Medicine, Instituto Português de Oncologia do Porto (IPO-Porto), 4200-072 Porto , Portugal; 2OncoPharma Research Group, Center for Health Technology and Services Research (CINTESIS), Rua Doutor Plácido da Costa, 4200-450 Porto, Portugal; nunovale@med.up.pt; 3Faculty of Medicine, University of Porto, Alameda Professor Hernâni Monteiro, 4200-319 Porto, Portugal; 4CINTESIS@RISE, Faculty of Medicine, University of Porto, Alameda Professor Hernâni Monteiro, 4200-319 Porto, Portugal; 5Department of Community Medicine, Health Information and Decision (MEDCIDS), Faculty of Medicine, University of Porto, Rua Doutor Plácido da Costa, 4200-450 Porto, Portugal; 6Cancer Epidemiology Group, Centro de Investigação do Instituto Português de Oncologia do Porto (CI-IPOP), 4200-072 Porto, Portugal; teresa.monjardino@ipoporto.min-saude.pt; 7Department of Anesthesiology, Centro Hospitalar Universitário de São João, Alameda Professor Hernâni Monteiro, 4200-319 Porto, Portugal; joanamourao@med.up.pt; 8Surgery and Physiology Department, Faculty of Medicine, University of Porto, Rua Doutor Plácido da Costa, 4200-450 Porto, Portugal

**Keywords:** lidocaine, intravenous, cytokines, inflammatory response, elective surgery, meta-analysis

## Abstract

There has recently been increasing evidence that the use of perioperative intravenous lidocaine infusion possesses analgesic, opioid-sparing and anti-inflammatory effects in surgical patients. Although opioid-sparing and analgesic properties have been strongly supported, the anti-inflammatory features are not well established in elective surgery. Therefore, the aim of this systematic review is to examine the effect of perioperative intravenous lidocaine infusion on postoperative anti-inflammatory status in patients undergoing elective surgery. A search strategy was created to identify suitable randomised clinical trials (RCTs) in PubMed, Scopus, Web of Science and Clinicaltrials.gov databases until January 2023. RCTs that evaluated the effect of intravenous lidocaine infusion, compared with placebo, on adult patients who underwent elective surgery, in inflammatory markers response were included. Exclusion criteria consisted of paediatric patients, animal studies, non-RCT methodology, intervention without intravenous lidocaine, inadequate control group, duplicated samples, ongoing studies and lack of any relevant clinical outcome measures. The following inflammatory markers—interleukin (IL)-6, tumour necrosis factor (TNF)-α, IL-1RA, IL-8, IL-10, C-reactive protein (CRP), IL-1, IL-1β, interferon (IFN)-γ, cortisol, IL-4, IL-17, high-mobility group protein B1 (HMGB1) and transforming growth factor (TGF)-β—were evaluated as outcomes in this review. A total of 21 studies, including 1254 patients, were identified. Intravenous lidocaine infusion significantly reduced the change from IL-6 baseline levels at the end of surgery compared to a placebo (standardised mean difference [SMD]: −0.647, 95% confidence interval [CI]: −1.034 to −0.260). Usage of lidocaine was associated with a significant reduction in other postoperative pro-inflammatory markers, such as TNF-α, IL-1RA, IL-8, IL-17, HMGB-1 and CRP. There was no significant difference in other markers, such as IL-10, IL-1β, IL-1, IFN-γ, IL-4, TGF-β and cortisol. This systematic review and meta-analysis provide support for the administration of perioperative intravenous lidocaine infusion as an anti-inflammatory strategy in elective surgery.

## 1. Introduction

Postoperative pain control is a major concern in anaesthesia, especially in the emerging field of fast-track surgery and in the presence of contra-indications for loco-regional anaesthesia. Opioid-free anaesthesia and analgesia is a major trend in Enhanced Recovery After Surgery (ERAS^®^) protocols.

ERAS protocols are the cornerstone of improved recovery after colorectal surgery. Their implementation leads to reduced morbidity and shorter hospital stays, while attenuating the surgical stress response. Multimodal analgesia is an important component of the ERAS protocols. The foundation of analgesia in ERAS pathways remains a multimodal approach, combining both regional anaesthesia and pharmacological agents, to reduce opioid dosing and its unwanted side effects, such as respiratory depression, ileus, nausea and vomiting, urinary retention, hyperalgesia, immune suppression, tolerance and postoperative delirium, among others [1].

In the last two decades, intravenous lidocaine infusion (IVLI) has been widely studied as a general anaesthesia adjuvant drug, and this approach has gained sustained interest in the medical literature, with the publication of several randomised controlled trials, descriptive reviews, systematic reviews and meta-analyses of the analgesic and opioid-sparing effect of IVLI in several clinical scenarios, from acute (perioperative included) to several chronic pain situations [2,3,4,5,6,7,8,9].

Despite huge evidence of efficacy and safety, IVLI for analgesia is not approved by the Food and Drug Administration (FDA) or the European Medicines Agency (EMA); intravenous administration is allowed only as an antiarrhythmic, with widespread resurgence of use that retains the indication ‘off-label’ [10]. To address this issue, an international consensus on the safety of IVLI was released in 2021 [11].

Lidocaine has been described as having analgesic, anti-hyperalgesic, antitumoral and immunomodulatory properties, suggesting that IVLI may provide significant postoperative benefits and opioid-sparing effects, as well as improves intraoperative parameters, with a relevant emphasis on cancer surgical patients [12,13,14,15,16,17,18,19,20,21,22,23,24,25].

So far, there is no comprehensive knowledge to obtain a conclusive finding on the anti-inflammatory effect of lidocaine. Thus, we undertook a systematic review and a meta-analysis of randomised controlled trials to evaluate the efficacy of perioperative IVLI as an anti-inflammatory by measuring inflammatory biomarkers as a postoperative outcome in patients undergoing all types of surgery requiring general anaesthesia, compared to placebo.

## 2. Materials and Methods

This systematic review was conducted according to the Preferred Reporting Items for Systematic Reviews and Meta-Analyses (PRISMA) statement [26]. The protocol for this systematic review and meta-analysis was registered in the International Prospective Register of Systematic Reviews (PROSPERO) on 22 January 2023 (registration number CRD42023393470).

### 2.1. Eligibility Criteria

Studies that evaluated the effect of IVLI, compared to placebo, on the response of inflammatory markers were included. We included papers that studied adult patients (18 years of age or older), independent of gender, who underwent any elective or urgent surgical procedure on any part of the body, and only if the procedure required general anaesthesia. Exclusively, randomised controlled trials (RCTs), regardless of their population size, were included.

Of interest were interventions using perioperative IVLI only or intravenous lidocaine associated with other analgesic agents, comparing the use of different pharmacological analgesic strategies (with and without intravenous lidocaine) and also with placebo groups. Studies that included more than two study arms, but had IVLI and placebo groups, were included, and only those groups pertinent to this systematic review were considered. The IVLI was mandatorily started (with or without an intravenous bolus) prior to the incision and continued until the end of surgery. Exclusion criteria comprised paediatric patients, animal studies, non-RCT methodology, intervention without intravenous lidocaine, inadequate control group, duplicate samples, ongoing studies and lack of any relevant clinical outcome measures.

### 2.2. Search Strategy

A literature search was conducted in the PubMed, Scopus, Web of Science and Clinicaltrials.gov databases for all available literature on the subject until 31 January 2023.

The search strategy with the full queries applied is presented in Appendix A. Bibliographic references of all studies that met the inclusion criteria were hand searched for any additional suitable reports to ensure study inclusion was as complete as possible. No limitations were placed on the data or language for inclusion.

### 2.3. Study Selection and Data Extraction Process

Interventional studies involving animals or humans, and other studies that require ethical approval, must list the authority that provided approval and the corresponding ethical approval code.

Two researchers (PC and IC) independently screened all titles and abstracts and selected the studies to include in the review according to the eligibility criteria. The articles included by title and abstract were then obtained, and the full text was read by the two authors, and it was decided if they met the inclusion criteria. Any discrepancies were solved by a tie-break of a third researcher (JM). The data extraction from the included studies was performed independently by both researchers (PC and IC). Any remaining disparities were discussed with the third researcher (JM).

From each eligible study, we collected information about the study (year of publication, study period, country, context and sample size), patient baseline demographics (age and gender distribution), and type of surgery, details of the lidocaine administration protocol (intraoperative bolus and continuous infusion doses, starting point, postoperative duration and stopping point), as well as the details of the placebo administered and the method and time of assessment of the selected inflammatory markers. The following inflammatory markers—interleukin (IL)-6, tumour necrosis factor (TNF)-α, IL-1RA, IL-8, IL-10, C-reactive protein (CRP), IL-1, IL-1β, interferon (IFN)-γ, cortisol, IL-4, IL-17, high-mobility group protein B1 (HMGB1) and transforming growth factor (TGF)-β—were evaluated as outcomes in this review.

Serum levels of inflammatory markers at different time points (before surgery, at the end of surgery, 12 h after surgery and 24 h postoperatively) in the treatment and placebo groups were obtained as the mean and standard deviation (SD). If there were any missing data, we contacted the relevant authors to obtain further information. If we succeeded, we included the data in the analyses.

We calculated missing SDs from standard errors (SEs) as described in the *Cochrane Handbook for Systematic Reviews of Interventions* [27]. If data were not reported as the mean with standard deviation, we estimated the sample mean and standard deviation from the sample size, median, first and third quartiles and interquartile range, as described by Wan et al., 2014 [28]. If data were reported only as the median with interquartile range, without the values of first and third quartiles, we considered the interquartile range as 1.35 SDs, if the distribution of the outcome is similar to the normal distribution, in accordance with the *Cochrane Handbook for Systematic Reviews of Interventions* [27].

Where results were available only in graphical format and the authors did not respond to the request for raw data, data were extracted in either direct or indirect form using PlotDigitizer [29].

### 2.4. Risk of Bias and Certainty Assessment

The risk of bias for each individual study was evaluated using the Cochrane Risk of Bias Tool [30], by two review authors independently (PC and IC). The five domains for individually randomised trials focus upon bias arising from the randomisation process, bias due to deviations from the intended interventions, bias due to missing outcome data, bias in the measurement of the outcome and bias in the selection of the reported result.

Certainty assessment was performed using GRADEpro GDT software to prepare the Summary of Findings table.

### 2.5. Statistical Analysis

Meta-analysis to summarise the effect of IVLI in inflammatory response was performed for each inflammatory marker with at least three studies using R software [31]. We analysed both mean (and SDs) concentrations, at specific time points, and concentration variations (from before surgery to end of surgery, from before surgery to 24 h post-surgery, from end of surgery to 24 h post-surgery) of serum inflammatory markers as outcomes. Pooled effect sizes were calculated using a random-effects approach with the restricted maximum likelihood method. Pooled effect sizes were expressed as standardised mean differences (SMDs) with a 95% CI.

Heterogeneity was evaluated by using I^2^ and Cochran’s Q test. Following the interpretation of Cochrane Handbook for Systematic Reviews [27], we classified heterogeneity as not important for I^2^ of 0% to 40%, as moderate for I^2^ of 30% to 60%, as substantial for I^2^ of 50% to 90% and as considerable for I^2^ of 75% to 100%. Substantial and considerable heterogeneity were investigated by subgroup analysis (setting and comparator/control intervention) [32]. The following subgroups, defined a priori, were examined: type of surgery, gender, publication year, location, data extraction method, time and dosing of lidocaine administration and risk of bias.

For robustness of the assessment, we performed a leave-one-out sensitivity analysis by removing one study at a time from the meta-analysis, to assess whether one study exerted a major impact on between-study heterogeneity. Pooled SMDs were recomputed, extracting these studies as necessary.

## 3. Results

### 3.1. Selection of Studies

The initial literature systematic search across four electronic databases identified a total of 519 studies according to the search strategy, as follows: PubMed (*n* = 134), Scopus (*n* = 222), Web of Science (*n* = 157) and Clinicaltrials.gov (*n* = 6). Among these, 179 records were removed because they were duplicates. A total of 340 studies were screened, and 303 of these were excluded because they did not meet the inclusion criteria. Cohen’s kappa of agreement was 88% and should be interpreted as a strong agreement between the 2 authors [33]. The remaining 36 studies were sought for retrieval, and the full studies were carefully screened, with a total of 21 studies included in this systematic review. The study selection process is outlined in the PRISMA diagram in Figure 1.

The 21 RCT studies included a total of 1254 patients. We summarised the characteristics of the included studies in Table 1.

The studies included were from 2006 onward.

### 3.2. Study Sample Characteristics

The overall sample size ranged from 16 participants [34] to 134 participants [35].

The proportion of male and female participants varied in the studies. In 6 trials, the proportion of female participants was more than 75% [36,37,38,39,40,41]. In 1 of the trials, male participants counted for more than 75% [42]. In 2 trials, there was an imbalance of the gender distribution between the experimental and control groups (>20%) [43,44]. We were unable to identify the gender distribution in one of the trials [45].

### 3.3. Surgical Procedures

In nine studies, open abdominal surgery was performed: abdominal hysterectomy [37,39,41], cholecystectomy [45], colorectal surgery [46,47,48], mixed major open abdominal procedures [35] and pelvi-abdominal cancer surgery [49]. In eight studies, video-assisted minimally invasive procedures were conducted: laparoscopic cholecystectomy [50,51], laparoscopic renal surgery [52], laparoscopic hysterectomy [40], laparoscopic gastroplasty [53], laparoscopic colorectal surgery [43], laparoscopic radical gastrectomy [54] and video-assisted thoracic surgery [38]. The remaining four studies looked at various other surgical procedures, such as supratentorial tumour resection [44], lumpectomy [34], thyroidectomy [36] and radical esophagectomy [42].
jcm-12-03772-t001_Table 1Table 1Characteristics of the studies.AuthorsYearCountryType of SurgeryASA Physical StatusSample SizeIntervention GroupControl GroupMarkers AssessedTime of AssessmentConclusionsAfzal et al. [45]2022Pakistanabdominal cholecystectomyNot reported (NR)80IV lidocaine bolus (2 mg/kg) followed by continuous infusion (1.5 mg/kg/h) intraoperativelyTreated likewise intervention group with NSIL-6 and IL-8End of surgery, 2 h postoperatively, 6 h postoperatively and 8 h postoperativelyIL-6 and IL-8 levels were significantly lower in IG than in CGZhao et al. [44]2022Chinasupratentorial tumour resectionI, II and III60IV lidocaine bolus (1.5 mg/kg) before induction of anaesthesia followed by continuous infusion (2 mg/kg/h) until the end of the surgeryTreated likewise intervention group with NSIL-6 and TNF-αPreoperatively and end of surgeryIL-6 and TNF-α levels were significantly lower in IG than in CGHou et al. [38]2021Chinavideo-assisted thoracic surgery for non-small-cell lung cancerI, II and III60IV lidocaine bolus (1.0 mg/kg) followed by continuous infusion (1.0 mg/kg/h) until the end of the surgeryTreated likewise intervention group with NSIL-17 and cortisol24 h postoperatively and at the time of post-anaesthesia care unit dischargeIL-17 and cortisol levels were significantly lower in IG than in CGXu et al. [40]2021Chinalaparoscopic hysterectomyI and II80IV lidocaine bolus (1.5 mg/kg) 10 min before induction of anaesthesia followed by continuous infusion (1.5 mg/kg/h) until 30 min before the end of the surgeryTreated likewise intervention group with NSIL-1, IL-6 and TNF-αBaseline, end of surgery, 2 h postoperatively and 24 h postoperativelyIL-1, IL-6 and TNF-α levels were not significantly lower in IG than in CGvan den Heuvel et al. [34]2020NetherlandslumpectomyI and II16IV lidocaine bolus (1.5 mg/kg) 10 min before induction of anaesthesia followed by continuous infusion (2 mg/kg/h) until 1 h postoperativelyTreated likewise intervention group with NSIL-1β, IL-1Ra, IL-6 and IL-10Before surgery and 4 h postoperativelyIL-1β, IL-1Ra, IL-6 and IL-10 levels were not significantly lower in IG than in CGOliveira et al. [53]2020Brazillaparoscopic gastroplastyII and III58IV lidocaine bolus (1.5 mg/kg) 5 min before induction of anaesthesia followed by continuous infusion (2 mg/kg/h) until the end of the skin sutureTreated likewise intervention group with NSIL-6Before surgery, 1 h and 5 h after the start of the surgery and 24 h postoperativelyIL-6 levels were not significantly lower in IG than in CGOrtiz et al. [51]2016Brazillaparoscopic cholecystectomyI and II43IV lidocaine bolus (1.5 mg/kg) at the start of the surgery followed by continuous infusion (3 mg/kg/h) until 1 h postoperativelyTreated likewise intervention group with NSIL-1, IL-6, IL-10, IFN-γ and TNF-αBefore surgery, 1 h and 5 h after the start of the surgery and 24 h postoperativelyIL-1, IL-6 IFN-γ and TNF-α levels were significantly lower in IG than in CG; IL-10 was significantly higher in IG than in CGWang et al. [39]2015Chinaradical hysterectomyI and II30Intravenous (IS) lidocaine bolus (1.5 mg/kg) 10 min before induction of anaesthesia followed by continuous infusion (1.5 mg/kg/h) until discharge from the operation roomTreated likewise intervention group with normal saline (NS)HMGB1, IFN-γ and IL-4Before surgery and 48 h postoperativelyHMBG1 levels were significantly lower in the intervention group (IG) than in the control group (CG); IFN-γ and IL-4 levels were not significantly lower in IG than in CGSridhar et al. [35]2014Indiaopen abdominal surgeriesI, II and III134IV lidocaine bolus (1.5 mg/kg) at intubation followed by continuous infusion (1.5 mg/kg/h) until 1 h postoperativelyTreated likewise intervention group with NSCRP and IL-6Before surgery, immediately after surgery and 24 h postoperativelyCRP levels were significantly lower in IG than in CG in the immediate postoperative period; IL-6 levels were significantly lower in IG than in CGWuethrich et al. [52]2012Switzerlandlaparoscopic renal surgeryI, II and III64IV lidocaine bolus (1.5 mg/kg) during induction of anaesthesia followed by continuous infusion (2 mg/kg/h) until the end of the surgery, followed by continuous infusion (1.3 mg/kg/h) until 24 h postoperativelyTreated likewise intervention group with NSCRP and cortisolBefore surgery, 24 h postoperatively and 48 h postoperativelyCRP and cortisol levels did not differ significantly between the two groupsElhafz et al. [43]2012Egyptlaparoscopic colorectal surgeryI, II and III18IV lidocaine infusion (2 mg/min if BW > 70 kg or 1 mg/min if BW < 70 kg) after induction of anaesthesiaTreated likewise intervention group with NSIL6, IL8 and TNF-αBefore surgery, 1 h, 24 h and 48 h postoperativelyIL-6, IL-8 and C3a levels were significantly lower in IG than in CG; TNF-α levels were not significantly lower in IG than in ICYardeni et al. [41]2009Israeltransabdominal hysterectomyI and II60IV lidocaine bolus (2 mg/kg) 20 min before the beginning of the surgery followed by continuous infusion (1.5 mg/kg/h) until the end of the surgeryTreated likewise intervention group with NSIL-1RA and IL-6Before surgery, 24 h postoperatively, 48 h postoperatively and 72 h postoperativelyIL-6 levels were significantly lower in IG than in CG; IL-1RA levels were not significantly lower in IG than in CGHerroeder et al. [47]2007Germanycolorectal surgeryI, II and III60IV lidocaine bolus (1.5 mg/kg) before induction of anaesthesia followed by continuous infusion (2 mg/min) until 4 h postoperativelyTreated likewise intervention group with NSIL-6, IL-8, IL-1β, TNF-α, IL-1RA, IL-10End of surgery, 2 h postoperatively and 24 h postoperativelyIL-6, IL-8 and IL-1RA levels were significantly lower in IG than in CG; TNF-α and IL-10 were not significantly lower in IG than in CGKuo et al. [46]2006Taiwancolonic surgeryI and II40IV lidocaine bolus (2 mg/kg) for 10 min followed by continuous infusion (3 mg/kg/h) until the end of the surgeryTreated likewise intervention group with NSIL-6, IL-8 and IL-1RABefore surgery, end of surgery, 12 h postoperatively and 24 h postoperativelyIL-6, IL-8 and IL-1ra levels were significantly lower in IG than in CGHamed et al. [48]2022Egyptmajor abdominal surgeryI and II54IV lidocaine bolus (1.5 mg/kg) followed by continuous infusion (2 mg/min) until 4 h postoperativelyTreated likewise intervention group with NSTNF-α and TGF-β4 h after recovery from anaesthesiaTNF-α levels were significantly lower in IC than in CG; TGF-β levels were significantly higher in IG than in CGHassan et al. [49]2022Egyptpelvi-abdominal cancer surgeryII and III36IV lidocaine bolus (1.5 mg/kg) followed by continuous infusion (1.5 mg/kg/h) until the end of the surgeryTreated likewise intervention group with NSIL-6 and TNF-αBefore surgery, end of surgery and 24 h postoperativelyIL-6 and TNF-α were significantly lower in IG than in CGSong et al. [50]2017Chinalaparoscopic cholecystectomyI, II and III71IV lidocaine bolus (1.5 mg/kg) 30 min before the skin incisions followed by a continuous infusion (2 mg/kg/h) until the end of the surgeryTreated likewise intervention group with NSIL-1RA, IL-6 and IL-8Before surgery, end of surgery and 12 h postoperativelyIL-6 and IL-8 were significantly lower in IG than in CG; IL-1RA was not significantly different between IG and CGOliveira et al. [37]2015BrazilhysterectomyI and II40IV lidocaine infusion (2 mg/kg/h) until the end of the surgeryTreated likewise intervention group with NSIL-6Before surgery, 5 h after the beginning of the surgery and 24 h postoperativelyIL-6 levels were not significantly different between IG and CGChoi et al. [36]2016South KoreathyroidectomyI, II and III56IV lidocaine bolus (1.5 mg/kg) followed by continuous infusion (2 mg/kg/h) until the end of the surgeryTreated likewise intervention group with NSCRPBefore surgery, 2 h postoperatively, 24 h postoperatively and 72 h postoperativelyCRP levels were significantly lower in IG than CGLv et al. [54]2021Chinalaparoscopic radical gastrectomyI, II and III90IV lidocaine bolus (1.5 mg/kg) followed by continuous infusion (1.5 mg/kg/h) until the end of the surgeryTreated likewise intervention group with NSIL-6, IL-10 and TNF-αBefore induction, 2 h after ventilation and at the end of surgery, 24 h postoperatively and 72 h postoperativelyIL-6, IL-10 and TNF-α levels were significantly lower in IG than in CGYuan et al. [42]2019Chinaradical esophagectomyII and III59IV lidocaine bolus (1.5 mg/kg) 5 min before induction of anaesthesia followed by continuous infusion (1.5 mg/kg/h) until 1 h postoperativelyTreated likewise intervention group with NSIL-6 and CRPBefore surgery, admission to the ICU, 12 h postoperatively and 36 h postoperativelyIL-6, IL-10 and TNF-α levels were significantly lower in IG than in CG


### 3.4. Study Conduct

We noted geographical variability among the studies. Seven studies were conducted in China [38,39,40,42,44,50,54], three in each of the countries Brazil [37,51,53] and Egypt [43,48,49] and one in each of the following countries: Pakistan [45], the Netherlands [34], India [35], Switzerland [52], Israel [41], Germany [47], Taiwan [46] and South Korea [36].

### 3.5. Details of Lidocaine Administration

Table 1 presents a summary of the details of lidocaine administration for each study.

In 19 studies, intravenous lidocaine administration was initiated with a bolus dose of 1 mg/kg to 2.5 mg/kg of body weight, with 1.5 mg/kg being the most common dose, used in 15 of the included studies [34,35,36,39,40,42,44,47,48,49,50,51,52,53,54]. In two studies, lidocaine administration was started as a continuous infusion without a bolus dose [37,43].

The dose of lidocaine infusion varied among studies from 1 mg/kg/h to 3 mg/kg/h. In 8 studies, the continuous infusion of lidocaine was delivered with a rate of <2 mg/kg/h [34,36,37,44,46,50,51,53], whereas an infusion rate of ≥2 mg/kg/h was applied in 12 other studies [35,38,39,40,41,42,43,45,47,48,49,54]. In 1 study, a higher infusion dose (≥2 mg/kg/h) was employed during the first study period, followed by continuous infusion (<2 mg/kg/h) during the second study period [52].

The continuous lidocaine infusion was terminated either at the end of the surgical procedure or with skin closure [36,37,38,40,41,44,46,49,50,53,54], 1 hour after the end of the surgery/skin closure [34,35,42,51], 4 hours postoperatively [47,48] and 24 h postoperatively [52]. Two studies did not report the exact time point for stopping the lidocaine infusion [43,45].

### 3.6. Risk of Bias

Seven studies were judged to be low risk in overall bias [34,38,40,44,49,52,53]. Thirteen studies were classified as having some concerns about overall bias [35,36,37,39,41,42,43,46,47,48,50,51,54], and one study was listed as high risk for overall bias [45]. The domain rated to have the highest risk of bias was bias in the selection of the reported result, mainly due to the absence of a registered trial. The risk of bias assessment is presented in Figure 2. The risk of bias assessment for each individual study is presented in Appendix A.

### 3.7. Outcomes

#### 3.7.1. IL-6

Overall, 16 studies evaluated the levels of the inflammatory marker IL-6 at different times among the studies. Fourteen of them included preoperative levels, while ten studies included immediate postoperative levels. The postoperative period analysed ranged from 1 to 72 h after surgery. Overall, 9 studies evaluated the period until 12 h after surgery, 11 studies analysed marker levels at 24 h after surgery and 4 studies included the period until 72 h after surgery. The inflammatory marker levels of IL-6 were significantly reduced in 75% of the studies [35,41,42,43,44,45,46,47,49,50,51,54]. The other studies (25%) found no differences between both groups [34,37,40,53].

A meta-analysis was calculated to evaluate this outcome and was conducted for RCTs that followed the lidocaine administration protocol and provided adequate statistical measures. IL-6 levels were analysed by computing effects based on the change from baseline at the end of the surgery and 12 h or 24 h after surgery.

A total of eight studies considered the change from IL-6 baseline levels at the end of the surgery, five in the open-surgery group [42,44,46,47,49] and three in the laparoscopic group [40,45,50,54]. Overall, IVLI was associated with significantly lower IL-6 levels by −0.647 standardised units (95% CI [−1.034; −0.260]). In those undergoing open surgery, a significant difference was also reported (−0.914 with a 95% CI [−1.401; −0.426]). However, no significant differences were observed in patients undergoing laparoscopic surgery (−0.261 with 95% CI [−0.684; 0.161]) (Figure 3).

There was substantial heterogeneity (I^2^ = 74%), which was investigated with subgroup analysis. The test for subgroup differences suggests that there is a statistically significant subgroup effect (*p* = 0.05), meaning that the type of surgery significantly modifies the effect of IVLI. However, there is substantial unexplained heterogeneity between the trials within each of these subgroups (open surgery: I^2^ = 69%; laparoscopic surgery: I^2^ = 64%). The impact of the following subgroups on the heterogeneity was investigated and was not statistically significant: gender, publication year, location, data extraction method, time and dosing of intravenous lidocaine administration and risk of bias.

The leave-one-out sensitivity analysis suggests that there is not one study exerting a major impact on between-study heterogeneity. As there were less than ten studies in this meta-analysis, a meta-regression was not considered, in accordance with the *Cochrane Handbook for Systematic Reviews of Interventions* [27].

Similar findings were observed in 9 studies that analysed the change of IL-6 levels from baseline at 24 h after surgery, 5 in the open group [41,42,45,46,47,49] and 4 in the laparoscopic group [40,50,53,54]. Overall, IVLI was associated with significantly lower IL-6 levels by −0.745 standardised units (95% CI [−1.248; −0.242]) and in those undergoing open surgery (−0.948 with a 95% CI [−1.564; −0.332]). However, no significant differences were observed in patients undergoing laparoscopic surgery (−0.497 with a 95% CI [−1.342; 0.348]) (Figure 4). There was also considerable heterogeneity (I^2^ = 85%) not fully explained by subgroup analysis with the different heterogeneity moderators assessed.

Likewise, no study was exerting a major impact on between-study heterogeneity, according to the leave-one-out sensitivity analysis, and no meta-regression was performed.

Publication bias was assessed by a funnel plot diagram. The funnel plot diagrams of the change from IL-6 baseline levels at the end of surgery and 24 h postoperatively were symmetrical and did not indicate the existence of publication bias (Figure 5 and Figure 6).

#### 3.7.2. TNF-α

For the inflammatory marker TNF-α, eight studies were evaluated, six of which measured preoperative levels, and the levels at the end of the surgery were obtained in seven studies. Additionally, postoperative levels were evaluated at various intervals ranging from 2 to 72 h. The levels of TNF-α were significantly reduced in 62.5% of the studies [44,48,49,51,54]. The other studies (37.5%) found no differences between both groups [40,43,47].

A meta-analysis was calculated to evaluate this outcome for RCTs that followed the lidocaine administration protocol and provided adequate statistical measures.

The TNF-α levels were analysed by computing effects based on the change from baseline at the end of the surgery and 24 h or 12 h after surgery.

A total of four studies considered the change from TNF-α baseline levels at the end of the surgery [40,44,47,54]. Overall, IVLI was associated with significantly lower TNF-α levels by −0.376 standardised units (95% CI [−0.609; −0.144]) (Figure 7). The heterogeneity was considered as not important (I^2^ = 0%). The leave-one-out sensitivity analysis suggested that one study [49] was exerting a major impact on between-study heterogeneity and was, therefore, not considered in this meta-analysis.

For the change of TNF-α levels from baseline at 12 h or 24 h after surgery, studied in 5 papers [40,47,49,51,54], IVLI was associated with lower TNF-α levels by −0.428 standardised units (95% CI [−1.211; 0.355]) (Figure 8). However, this difference was not significant. There was also considerable heterogeneity (I^2^ = 90%). Due to the small number of studies included, it is unlikely that an investigation of heterogeneity will produce useful findings [27].

#### 3.7.3. IL-1RA

In general, 5 studies evaluated the levels of the inflammatory marker IL-1RA at different times, including preoperative levels, at the end of surgery and during the postoperative period, up to 72 h after surgery.

Most of the studies did not report any statistically significant difference in the levels of IL-1RA among the groups studied [34,41,50]. However, 40% of the studies reported statistically significant reductions in IL-1RA levels in the IVLI group [46,47].

A meta-analysis was computed to evaluate this outcome for RCTs that followed the lidocaine administration protocol and provided adequate statistical measures (Figure 9). This meta-analysis analysed the change of IL-1RA levels from baseline at 24 h after surgery in 3 studies [41,46,47]. Overall, IVLI was associated with significantly lower IL-1RA levels by −0.553 standardised units (95% CI [−0.901; −0.205]). The heterogeneity was classified as not important (I^2^ = 19%). The leave-one-out sensitivity analysis suggested that one study [50] was exerting a major impact on between-study heterogeneity and was, therefore, not considered in this meta-analysis.

Due to the low number of studies reporting a change from IL-1RA baseline levels at the end of the surgery, no meta-analysis was performed for this period.

#### 3.7.4. IL-8

In total, five studies analysed the levels of the inflammatory marker IL-8 as a result. Three of them included preoperative levels, and the levels at the end of the surgery were obtained in three studies. Furthermore, various intervals in the postoperative period, ranging from 2 to 48 h after surgery, were included.

All studies reported a statistically significant difference in the levels of the inflammatory marker IL-8 in the IVLI group compared to the control [43,45,46,47,50].

A meta-analysis was calculated to evaluate this outcome for RCTs that followed the lidocaine administration protocol and provided adequate statistical measures (Figure 10). The IL-8 levels were analysed at the end of the surgery [46,47,50] due to the insufficient number of studies reporting a change from IL-8 baseline levels. In general, IVLI was associated with lower levels of IL-8 by −0.792 standardised units (95% CI [−1.600; 0.016]). However, this difference was not statistically significant. There was also substantial heterogeneity (I^2^ = 82%).

#### 3.7.5. IL-10

In total, four studies analysed the levels of the inflammatory marker IL-10 as an outcome. Two of them included preoperative levels, and the levels at the end of the surgery were obtained in three studies. Furthermore, various intervals in the postoperative period, ranging from 2 to 72 h after surgery, were included.

One of the studies reported that the levels of IL-10 were reduced in the experimental group [54]. Two of the studies reported that there was no significant difference between both groups [34,47], and one of the studies reported significantly higher levels of IL-10 in the experimental group [51].

A meta-analysis was performed to analyse this result for RCTs that followed the lidocaine administration protocol and provided adequate statistical measures (Figure 11). IL-10 levels were analysed at the end of the surgery [47,51,54] due to the insufficient number of studies reporting a change from IL-10 baseline levels. Overall, IVLI was associated with reduced IL-10 levels by −0.624 standardised units (95% CI [−1.263, 0.015]). However, there was substantial heterogeneity (I^2^ = 80%).

#### 3.7.6. CRP

Four studies analysed the levels of CRP as a result. All of them measured CRP levels before surgery, but only one did in the immediate period after surgery. Furthermore, various intervals in the postoperative period, ranging from 2 to 72 h after surgery, were included.

In three of the studies, the IVLI group was reported to be associated with significantly reduced levels of CRP [35,36,42]. In 1 of the studies (25%), there were reported to be no significant differences between the groups [52].

The CRP levels were mainly reported at different times in the studies. Due to this heterogeneity in the reporting style, no meta-analysis was performed for this outcome.

#### 3.7.7. IL-1

The levels of the inflammatory marker IL-1 were reported by two studies. Only 1 of them reported the levels before surgery, and both measured the levels after surgery and at 24 h postoperatively. A study reported significantly lower IL-1 levels in the lidocaine [51] group, but the other one did not find any significant conclusions [40].

Due to the low number of studies reporting IL-1 levels, no meta-analysis was performed for this outcome.

#### 3.7.8. IL-1β

Two articles studied the levels of IL-1β as an outcome. However, both studies could not find significant differences between the groups [34,47].

Due to the low number of studies reporting IL-1β levels, no meta-analysis was performed for this outcome.

#### 3.7.9. IFN-γ

The levels of the inflammatory marker IFN-γ were reported by two studies. A study reported significantly lower IFN-γ levels in the lidocaine group [51], but the other one did not find any significant difference between the groups [39].

#### 3.7.10. Cortisol

The levels of cortisol were reported by two studies. A study reported significantly lower cortisol levels in the lidocaine group [38], but the other study did not find any significant differences between the groups [52].

#### 3.7.11. IL-4, IL-17, HMGB1 and TGF-β

The levels of IL-4, IL-17, HMGB1 and TGF-β were only reported by one study each. These studies reported significantly reduced levels of IL-17 and HMGB1 in the experimental group [38,39]. However, IL-4 levels were not significantly different when both groups were compared [39]. Furthermore, the levels of TGF-β were reported to be significantly superior in the lidocaine group [48].

### 3.8. GRADE Assessment

The quality of the evidence of the main outcomes in the present meta-analysis was evaluated with the GRADE assessment, described in Table 2.

The recommendation level of evidence is classified in the following four categories: High—We are very confident that the true effect lies close to that of the estimate of the effect; Moderate—We are moderately confident in the effect estimate: The true effect is likely to be close to the estimate of the effect, but there is a possibility that it is substantially different; Low—Our confidence in the effect estimate is limited: The true effect may be substantially different from the estimate of the effect; Very low—We have very little confidence in the effect estimate: The true effect is likely to be substantially different from the estimate of the effect.

## 4. Discussion

This is the first meta-analysis to compare the efficacy of perioperative intravenous lidocaine with a placebo for its anti-inflammatory effect, in patients undergoing elective surgery. The most important finding of the present meta-analysis was that intravenous lidocaine was associated with a significant reduction of postoperative inflammatory markers.

The analgesic and anti-hyperalgesic effects of systemic lidocaine have been recognised for almost 70 years [24,55]. Previous studies have demonstrated its analgesic properties in a range of surgical specialties, reported extensively in a recent study by Weibel et al. [56], especially in colorectal and hepatobiliary surgery, as described in recent meta-analyses by Rollins et al. [8] and Li et al. [7], respectively.

In in vitro settlements, lidocaine has been shown to inhibit cancer cell behaviour and exerts beneficial effects on components of the inflammatory and immune responses that are known to affect cancer biology [19,20,22].

The exact mechanism of the effect of the anti-inflammatory properties of lidocaine still remains unclear. Lidocaine can inhibit leucocyte activation, adhesion and migration. It also protects cells from inflammation through the reduction of neutrophil adhesion and inhibition of the release of superoxide anions and also by blocking the release of inflammatory mediators, such as interleukins, IFN-γ, cortisol, HMGB1, TNF-α and TGF-β, in in vitro and in vivo studies. Furthermore, it also inhibits prostaglandin biosynthesis and release, which is another possible explanation for its powerful anti-nociceptive and anti-inflammatory actions [55].

Interleukins can be produced by all immune system cells, epithelial cells, fibroblasts and tumours. Their profile is dependent on the type of cells and immune response, which lays to their secretion. For what concerns this paper, IL-1, IL-2, IL-6, IL-8 and IL-17 are pro-inflammatory, while IL-4 and IL-10 are immunosuppressant or immunomodulatory, depending on the inflammation trigger. In a surgical aggression scenario, both have negative effects on immune function [57,58].

The findings of this study demonstrated that the levels of IL-6 were significantly reduced at the end of surgery and 24 h postoperatively in patients undergoing elective surgery, particularly in those undergoing open surgery, when treated with perioperative intravenous lidocaine infusion. This result supports the hypothesis that lidocaine has a significant anti-inflammatory effect, as IL-6 is a pro-inflammatory cytokine that is associated with surgical trauma.

Furthermore, a meta-analysis of the available literature revealed that intravenous lidocaine infusion significantly reduces IL-1RA levels at 24 h after surgery. However, no significant reduction in IL-8 and IL-10 levels was observed in the meta-analysis. Nonetheless, most studies included in the analysis reported a decrease in their levels, suggesting that lidocaine may have some anti-inflammatory effects on these cytokines, as well. Similar findings were detected with CRP.

TNF-α has a double role as a pro-inflammatory mediator by initiating a strong inflammatory response, as well as an immunosuppressive mediator, by limiting the extent and duration of inflammatory processes [58]. IVLI was associated with a significant reduction of TNF-α levels at the end of the surgery. However, at 24 h postoperatively, this difference was not significant.

Likewise, TGF-β balances pro-inflammatory and anti-inflammatory effects by decreasing the cellular growth of almost all immune cell precursors and acts in the regulation of the differentiation of several T-helper cell subsets [58]. Finally, HMGB1 acts as a damage-associated molecular pattern molecule, regulating inflammation and immune responses through different receptors or direct uptake [59]. Other biomarkers, such as IL-1β, IFN-γ, IL-4, IL-17 and cortisol, have also been implicated in the inflammatory response to surgical aggression. However, there are currently too limited data on the effect of perioperative lidocaine infusion on the levels of these inflammatory mediators to draw conclusions on its effect.

Thus, this systematic review suggests that lidocaine indeed has intrinsic anti-inflammatory activity, measured by specific pro-inflammatory markers, such as IL-1, IL-6, IL-8, TNF- α and PCR, as well immunomodulatory activity, as evidenced by IL-1R and IL-10, with statistical significance, despite high heterogeneity. Pro-inflammatory activity enhances inflammation by recruiting effector immune cells, altering vascular permeability and promoting tissue injuring, by a positive feedback loop; IL-1 and IL-6 promote cyclooxygenase 2 (COX) upregulation. Some cytokines and neuro-hormonal responses induce immunosuppression, blocking effector immune cells function by regulatory T and B cells, such as IL-1RA, IL-4 and IL-10 by a negative feedback loop. Immunosuppression can lead to infection, postoperative morbidity and cancer persistence, recurrence and metastasis. IVLI diminished both pro-inflammatory and immunosuppressive markers levels.

Classical anti-inflammatory drugs are steroids (such as dexamethasone and prednisolone) and non-steroids (such as aspirin, diclofenac, ibuprofen, oxicams and coxibes) that target arachidonic acid production and COXs (1 and 2) function. Many patients have relative and absolute contra-indications to non-steroidal anti-inflammatory drugs (NSAIDs) due to haemorrhagic complications, kidney disease or thrombotic conditions. From our study conclusions, we can strongly suggest that open-surgery patients (not eligible for locoregional anaesthesia/analgesia or NSAIDs) are the ones who will benefit the most from IVLI for perioperative analgesia. By extrapolation from literature reviews, surgical oncologic patients are those who could benefit more by adding lidocaine to a multimodal analgesic regimen [60].

However, it is important to note that this study had some limitations that warrant discussion. One limitation is the relatively small number of studies included in the meta-analysis for each outcome, which may have impacted the ability to detect significant differences between the intervention and control groups.

Additionally, as previously described, there was some variability among studies in the details of lidocaine administration, regarding its dose and duration, and in the range of surgical specialties conducted, which may have contributed to the heterogeneity observed in the meta-analysis.

Overall, while the current study provides valuable insights into the anti-inflammatory effect of perioperative intravenous lidocaine infusion, the limitations of the study highlight the need for further research to fully understand the impact of lidocaine on the inflammatory response to surgical aggression. Likewise, the evidence quality for each outcome is considered to be moderate, which means that further research is likely to have an important impact on our confidence in the estimate of the effect and may change the estimate.

From the articles eligible for our systematic review with meta-analysis, besides inflammatory markers’ behaviour under IVLI that was significantly improved, postoperative pain, nausea and vomiting and paralytic ileus were also measured and improved by IVLI, but were not chosen outcomes in our study. We suggest in the future to undertake a systematic review focusing on the clinical significance of decreased inflammation, targeting outcomes such as surgical site/wound infection, postoperative sepsis, anastomosis dehiscence, cardiorespiratory morbidity, cancer recurrence and metastasis.

## 5. Conclusions

There are several medicines targeting ILs and their receptors, aiming to enhance inflammation, control cachexia or induce immunosuppression, depending on the disease to treat. All these ILs-targeting drugs are promising, although very expensive, with inconsistent clinical results and not devoid of serious undesirable side effects [61].

Lidocaine is a well-known, cheap, safe and easily available drug that can induce an immune response towards a desirable equilibrium between surgical inflammatory response and immunomodulation, as demonstrated in this study.

Lidocaine’s mechanism of action does not seem to involve cyclooxygenases (COX 1 and 2) function inhibition, so, it could also be a resourceful alternative to NSAIDs in patients with kidney, heart and cerebrovascular disease, as well as bleeding disorders and asthma.

IVLI could be a powerful strategy to mitigate the negative effects caused by surgical aggression on the immune function, but remains targeted as ‘off-label’. More exhaustive and larger sample studies are needed to reinforce its positive impact, and, for that, it is crucial to obtain its approval as an analgesic and anti-inflammatory by intravenous administration, as a clinical indication, from international or national drug regulatory entities.

## Figures and Tables

**Figure 1 jcm-12-03772-f001:**
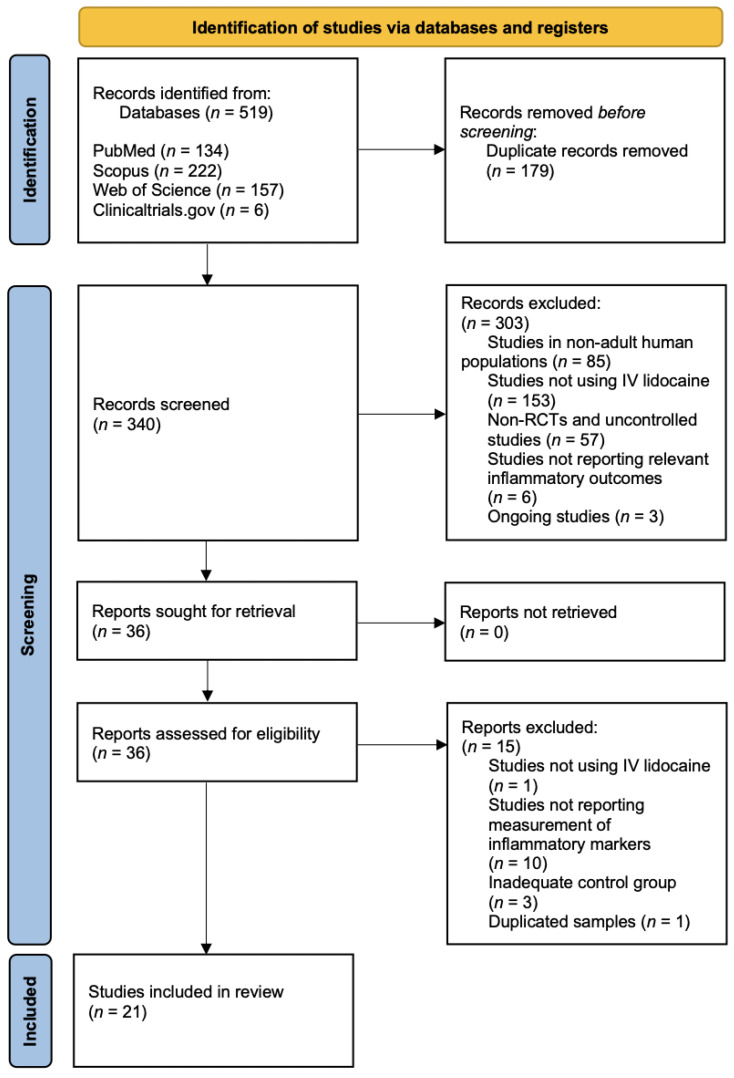
Study selection process.

**Figure 2 jcm-12-03772-f002:**
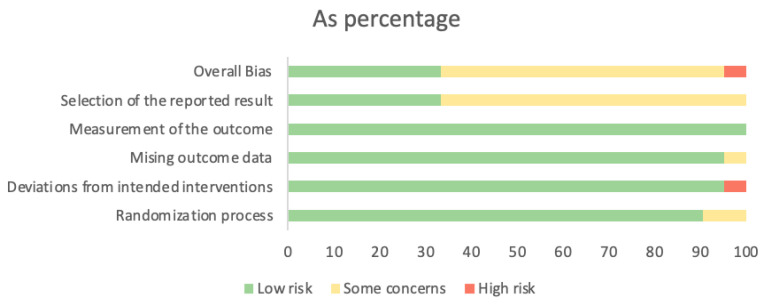
Risk of bias graph: review authors’ judgments about each risk of bias item presented as percentages.

**Figure 3 jcm-12-03772-f003:**
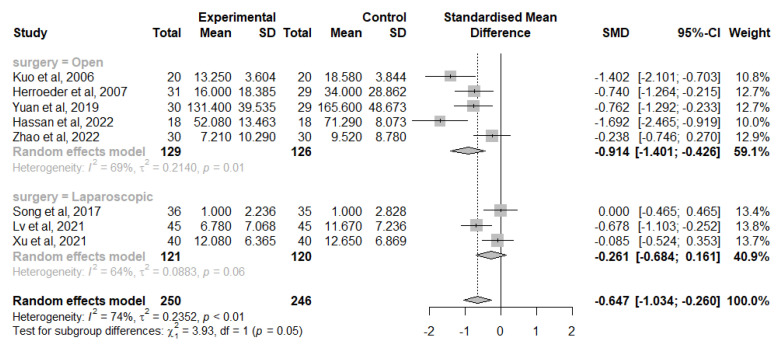
Forest plot of comparison: change from IL-6 baseline levels at the end of the surgery [40,42,44,46,47,49,50,54].

**Figure 4 jcm-12-03772-f004:**
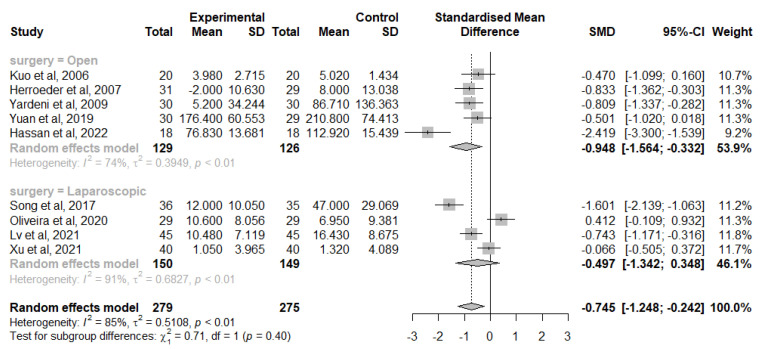
Forest plot of comparison: change from IL-6 baseline levels at 24 h after surgery [40,41,42,46,47,50,53,54].

**Figure 5 jcm-12-03772-f005:**
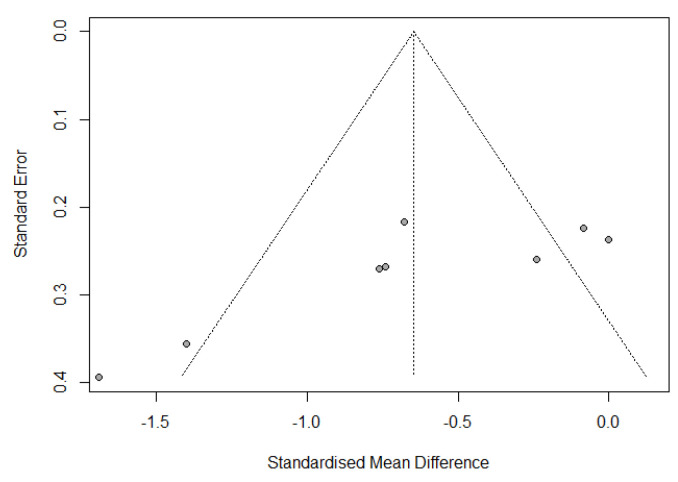
Funnel plot of change from IL-6 baseline levels at the end of the surgery.

**Figure 6 jcm-12-03772-f006:**
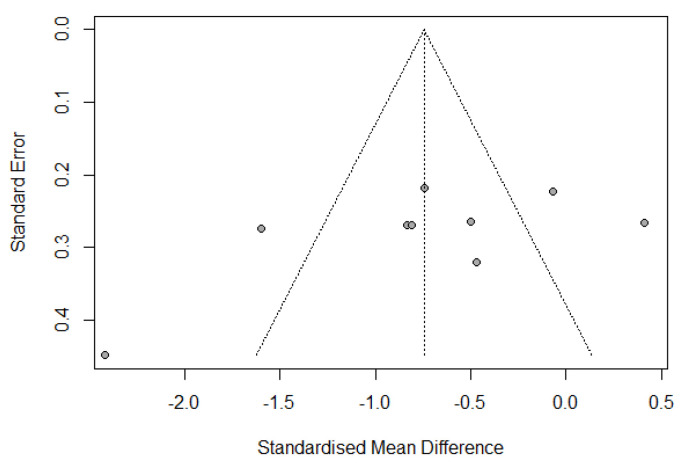
Funnel plot of change from IL-6 baseline levels at 24 h postoperatively.

**Figure 7 jcm-12-03772-f007:**
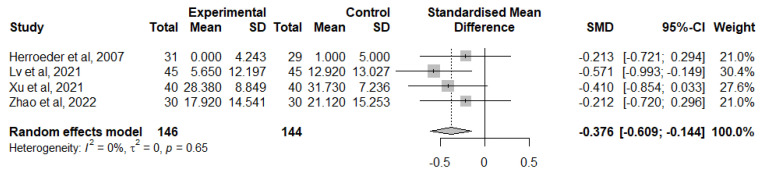
Forest plot of comparison: change from TNF-α baseline levels at the end of the surgery [40,44,47,54].

**Figure 8 jcm-12-03772-f008:**
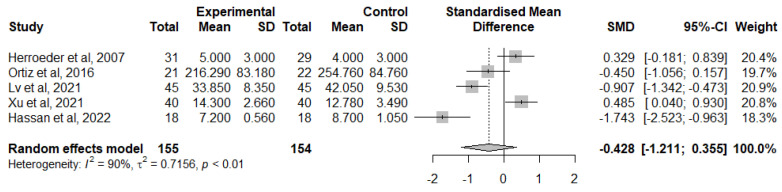
Forest plot of comparison: change from TNF-α baseline levels 24 h after surgery [40,47,49,51,54].

**Figure 9 jcm-12-03772-f009:**
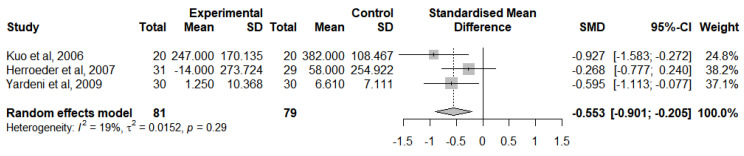
Forest plot of comparison: IL-1RA levels at 24 h after surgery [41,46,47].

**Figure 10 jcm-12-03772-f010:**
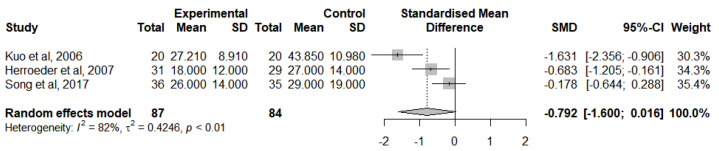
Forest plot of comparison: IL-8 levels at the end of the surgery [46,47,50].

**Figure 11 jcm-12-03772-f011:**
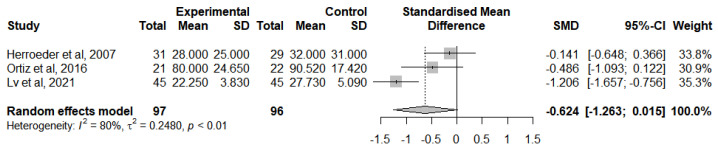
Forest plot of comparison: IL-10 levels at the end of the surgery [47,51,54].

**Table 2 jcm-12-03772-t002:** Summary of Findings with GRADE Assessment Patient or population: patients undergoing elective surgery. Setting: inpatients. Intervention: perioperative intravenous lidocaine infusion. Comparison: normal saline.

Outcomes	№ of Participants (Studies)	Certainty of the Evidence (GRADE)	Relative Effect (95% CI)	Anticipated Absolute Effects	
				Risk with Control	Risk Difference with Intravenous Lidocaine Infusion
change from IL-6 baseline levels at the end of the surgery	496 (8 RCTs)	Moderate ^a^	-	-	SMD 0.647 SD lower (1.034 lower to 0.26 lower)
change from IL-6 baseline levels at 24 h after surgery	554 (9 RCTs)	Moderate ^a^	-	-	SMD 0.745 SD lower (1.248 lower to 0.242 lower)
change from TNF-α baseline levels at the end of the surgery	290 (4 RCTs)	Moderate ^b^	-	-	SMD 0.376 SD lower (0.609 lower to 0.144 lower)
change from TNF-α baseline levels 24 h after surgery	309 (5 RCTs)	Low ^a,b^	-	-	SMD 0.428 SD lower (1.211 lower to 0.355 higher)
IL-1RA levels at 24 h after surgery	160 (3 RCTs)	Moderate ^b^	-	-	SMD 0.553 SD lower (0.901 lower to 0.205 lower)
IL-8 levels at the end of surgery	171 (3 RCTs)	Low ^a,b^	-	-	SMD 0.792 SD lower (1.6 lower to 0.016 higher)
IL-10 levels at the end of surgery	193 (3 RCTs)	Low ^a,b^	-	-	SMD 0.624 SD lower (1.263 lower to 0.015 higher)

The risk in the intervention group (and its 95% Confidence Interval) is based on the assumed risk in the comparison group and the relative effect of the intervention (and its 95% CI). CI, Confidence Interval; SMD, standardized mean difference, SD standard deviation; GRADE Working Group grades of evidence; High certainty, We are very confident that the true effect lies close to that of the estimate of the effect; Moderate certainty, We are moderately confident in the effect estimate: The true effect is likely to be close to the estimate of the effect, but there is a possibility that it is substantially different; Low certainty: Our confidence in the effect estimate is limited: The true effect may be substantially different from the estimate of the effect; Very low certainty: We have very little confidence in the effect estimate: The true effect is likely to be substantially different from the estimate of the effect. Explanations: ^a^ There was substantial heterogeneity, not fully explained by subgroup analysis. ^b^ Few events and wide confidence intervals.

## Data Availability

The data presented in this study are available on request from the corresponding author.

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
