# Peer review of "Systemic Anti-Inflammatory Effects of Intravenous Lidocaine in Surgical Patients: A Systematic Review and Meta-Analysis"

_jcm, 2023, doi:10.3390/jcm12113772_

Round 1

Reviewer 1 Report

In this manuscript, the authors performed studies to evaluating the anti-inflammatory  efficacy of IVLI in surgical patients. Overall, this work is well conducted. However, this meta-analysis does not clearly answer the question that which surgeries are the most suitable for use IVLI. In addition, it is difficult to interpret the significance of partial inflammatory cytokines alteration due to the variability of the disease and individual heterogeneity.

Author Response

Point 1: In this manuscript, the authors performed studies to evaluating the anti-inflammarory efficacy of IVL in surgical patients. However, this meta-analysis does not clearly answer the question that wich surgeries are the most suitable for use IVLI. In addition, i tis difficult to interpret the significance of partial inflammatory cytokines due to variability of the disease and individual heterogeneity.

Response to Point 1:

We agree that our meta-analysis included a short number of surgeries, (4 hysterectomy, 3 cholecystectomy, 2 colo-rectal surgery, 1 renal surgery, 1 gastroplasty, 1 gastrectomy, 1 mixed abdominal surgery, 1 pelvi-abdominal not colonic surgery, 1 thoracic surgery, 1 thyroidectomy, 1 lumpectomy, 1 esophagectomy, 1 brain surgery), but all of them are suitable for IVLI use. We opted to divide them in open and videoassisted, and could conclude with more strenght of evidence that the most suited is open surgery, with statistical significance for the outcomes  studied, despite of high heterogeneity. From what we searched from literature, perioperative intravenous lidocaine has anti-inflammatory effects in clinical concentrations, within therapeutical (sub-toxic) serum levels, and could be used in all surgical patients as a multimodal analgesic approach to reduce opioid use.

Helas, we have indeed limitations in our study, the most obvious is the reduced number of articles included (19), and the high heterogeneity. We tried to approach this issue seriously, including only RCT’s, respecting scrupulously inclusion/exclusion criteria and drug regimen protocols; we also used validated statistically tools to deal with that problem. We came to the conclusion that heterogeneity was independent of our study protocol, attributable probably to population characteristics, the variability of surgical techniques, and to the reduced number of studies included. As for “partial inflammatory cytokines”, we adressed all inflammatory markers available in our query search, and analized those studies with similar citokines profile.

We need more studies in the future contempling specific surgeries and a wider profile of inflammatory markers.

Reviewer 2 Report

Thank you for the opportunity to review this manuscript. The authors have conducted a thorough and excellent systematic review and meta-analysis. I have a few suggestions:

1) The suggested clinical utility of these discussions has not been discussed.

2) While statistical significance may have been achieved, were the authors able to investigate any clinical outcomes that were significantly improved?

3) In your supplementary material, please indicate all the values that were imputed using existing data.

4) Perhaps the authors could perform subgroup analyses by specialty if they have enough studies.

5) Why were articles excluded if they didn't include "relevant clinical outcome measures" as your review does not look at clinical outcome measures in the first place.

Author Response

Point 1: The suggested clinical utility of these discussion has not been discussed.

Response to Point 1:

To our knowledge, this is the first systematic review with meta-analysis that suggests lidocaine has indeed intrinsic anti-inflammatory activity, measured by specific pro-inflammatory markers as IL-1, IL-6, IL-8, TNF-alfa, PCR, as well immunomodulatory activity, as evidenced by IL-1R and IL-10, with statistical significance, despite high heterogeneity. Pro-inflammatory activity enhances inflammation, by recruiting effector immune cells, altering vascular permeability and promoting tissue injuring, by a positive feedback loop; IL-1 and IL-6 promote COX2 up-regulation. Some citokines and neuro-hormonal responses induce immunossupression, blocking effector immune cells function by regulatory T and B cells, as IL-1R, IL-4, IL-10, by a negative feedback loop. Immunossupression can lead to infection, postoperative morbidity and cancer persistance, recurrance and metastasis. IVL diminuished both pro-inflammatory as imunossupresive markers levels.

Classical anti-inflammatory drugs are steroid (dexamethasone, prednisolone) and non-steroid (aspirin, diclofenac ibuprofen, oxicams, coxibes, …); they target arachidonic acid production and COXes (1 and 2) function. Many patients have relative and absolute contra-indications to NSAID’s, due to hemorragic complications, kidney disease or thrombotic conditions. From our study conclusions, we can strongly suggest that open surgery patients (not eligible for locorregional anesthesia/analgesia or NSAID’s) are the ones who will benefit the most from IVL for perioperative analgesia. By extrapolation from literature reviews, surgical oncologic patients are those who could benefit more by adding lidocaine to a multimodal analgesic regimen.

Point 2: While statistical significance may have been achieved, were the authors able to investigate any clinical outcomes that were significantly improved?

Response to Point 2:

We agree with the reviewer that understanding the role of peri-operative intravenous lidocaine infusion in clinical outcomes would be of utmost importance. This review represents a first attempt to summarize the effect of IV lidocaine in inflammatory response and we specifically selected studies reporting measurement of inflammatory markers.

From the articles elegible to our systematic review with meta-analysis, besides inflammatory markers behavior under IVLI that was significantly improved, also postoperative pain, nausea and vomiting and paralitic ileus were also measured and also improved by IVL, but were not  chosen outcomes to our study. We suggest in the future to undertake a systematic review focusing on the clinical significance of decreased inflammation, targeting outcomes as surgical place/wound infection, postoperative sepsis, anastomosis dehiscence, cardiorrespiratory morbidity, cancer recurrence and metastasis.

Point 3:  In your supplementary material, please indicate all the values that were imputed using existing data.

Response to Point 3:

All values imputed are on our forest plot graphics and in our database, that will be provided at your request. 

Point 4:  Perhaps the authors could perform subgroup analyses by specialty if they have enough studies.

Response to Point 4:

Specialty means type of surgery? Unfortunately, it was not possible to perform this analysis due to short number of studies (4 hysterectomy, 3 cholecystectomy, 2 colo-rectal surgery, 1 renal surgery, 1 gastroplasty, 1 gastrectomy, 1 mixed abdominal surgery, 1 pelvi-abdominal not colonic surgery, 1 thoracic surgery, 1 thyroidectomy, 1 lumpectomy, 1 esophagectomy, 1 brain surgery). For that reason, we opted to separate surgical procedures in open/videoassisted,  with statistical significance in the results.

Point 5: Why were articles excluded if they didn't include "relevant clinical outcome measures" as your review does not look at clinical outcome measures in the first place.

Response to Point 5:

You are absolutely right, we agree with your suggestion, as we measured inflammatory markers, not relevant clinical outcomes. We will proceed to alterations in the manuscript.